# Identification and Validation of a Prognostic Signature Derived from the Cancer Stem Cells for Oral Squamous Cell Carcinoma

**DOI:** 10.3390/ijms25021031

**Published:** 2024-01-14

**Authors:** Mingxuan Shi, Ke Huang, Jiaqi Wei, Shiqi Wang, Weijia Yang, Huihui Wang, Yi Li

**Affiliations:** 1Key Laboratory of Dental Maxillofacial Reconstruction and Biological Intelligence Manufacturing, School of Stomatology, Lanzhou University, Lanzhou 730030, China; shimx2021@lzu.edu.cn (M.S.); huangk18@lzu.edu.cn (K.H.); 220220929560@lzu.edu.cn (J.W.); wangshq2020@lzu.edu.cn (S.W.); yweijia2023@lzu.edu.cn (W.Y.); 2Key Laboratory of Preclinical Study for New Drugs of Gansu Province, School of Basic Medical Sciences, Lanzhou University, Lanzhou 730030, China

**Keywords:** oral squamous cell carcinoma, cancer stem cells, prognosis signature, RNA-sequencing, *ADM*, *POLR1D*

## Abstract

The progression and metastasis of oral squamous cell carcinoma (OSCC) are highly influenced by cancer stem cells (CSCs) due to their unique self-renewal and plasticity. In this study, data were obtained from a single-cell RNA-sequencing dataset (GSE172577) in the GEO database, and LASSO-Cox regression analysis was performed on 1344 CSCs-related genes to establish a six-gene prognostic signature (6-GPS) consisting of *ADM*, *POLR1D*, *PTGR1*, *RPL35A*, *PGK1*, and *P4HA1*. High-risk scores were significantly associated with unfavorable survival outcomes, and these features were thoroughly validated in the ICGC. The results of nomograms, calibration plots, and ROC curves confirmed the good prognostic accuracy of 6-GPS for OSCC. Additionally, the knockdown of *ADM* or *POLR1D* genes may significantly inhibit the proliferation, migration, and invasion of OSCC cells through the JAK/HIF-1 pathway. Furthermore, cell-cycle arrest occurred in the G1 phase by suppressing Cyclin D1. In summary, 6-GPS may play a crucial role in the occurrence and development of OSCC and has the potential to be developed further as a diagnostic, therapeutic, and prognostic tool for OSCC.

## 1. Introduction

Despite advancements in science, oral cancer continues to have a high mortality rate in humans [1,2]. More than 90% of oral malignancies, according to histopathology, are caused by oral squamous cell carcinoma (OSCC) [3,4]. In 2020, 377,713 new cases of oral cancer were reported worldwide, along with 177,757 new fatalities [5]. The traditional techniques of treating OSCC include surgery, radiation, chemotherapy, and a combined approach based on the three [6]. Gene therapy, immunotherapy, hyperthermia, and photodynamic treatment used in conjunction with conventional therapy have helped OSCC patients’ therapeutic outcomes in recent years [7,8,9,10], but their 5-year overall survival rate (OS) is still only 50% to 64% [11]. The recurrence rate of OSCC is between 18% and 76%, even in individuals who receive the usual mix of surgical and non-surgical treatments [12]. The elucidation of molecular mechanisms driving OSCC occurrence is essential, as it holds the promise of expediting the advancement of personalized therapeutic approaches.

Recent advances have highlighted the significant role of cancer stem cells (CSCs) with self-renewal and differentiation capabilities in the occurrence and development of OSCC [13,14]. Additionally, CSCs are strongly linked to poor prognostic outcomes and cancer recurrence in OSCC due to their various biological differences from non-CSCs, including resistance to chemotherapy and radiotherapy [15,16], avoidance of induced cell death [17], and dormancy [18]. Reports indicate that certain CSCs express proteins similar to embryonic stem cells, such as *OCT4*, *NANOG*, and *SOX2*, which serve as key regulators for self-renewal and maintenance in undifferentiated stem cell populations [19,20]. As for potential CSCs, indicators in *OSCC*, *CD44*, *CD24*, *CD133*, *Oct-4*, *Bmi-1*, *Musashi-1*, and *ALDH* have also been mentioned [21,22]. However, currently, there is a lack of reliable biomarkers available for the accurate isolation and characterization of CSCs in OSCC. Therefore, the identification of novel and improved CSC markers that comprehensively correlate with known cancer progression alterations in OSCC appears to be necessary.

RNA sequencing (RNA-seq) is a high-throughput method for determining the sequence of RNA molecules in the transcriptome. Combining bulk RNA-seq and Single-cell RNA-seq (scRNA-seq) not only provides comprehensive information about gene expression but also aids in understanding the cellular genetic, transcriptional, and epigenetic changes during the progression of OSCC [23]. In this study, we obtained a scRNA-seq dataset from the Gene Expression Omnibus (GEO) and bulk RNA-seq data for OSCC from the International Cancer Genome Consortium (ICGC). Using bioinformatics tools, we analyzed and identified the cancer stem cell marker genes (CSCMGs) of OSCC and further constructed and validated a six-gene prognostic signature (6-GPS, consisting of Adrenomedullin (*ADM*), RNA polymerase 1 subunit D (*POLR1D*), Prostaglandin reductase 1 (*PTGR1*), L35a ribosomal protein (*RPL35A*), Phosphoglycerate kinase 1 (*PGK1*) and Prolyl 4-hydroxylase subunit alpha 1 (*P4HA1*)) for OSCC. Subsequently, we constructed nomograms for the prediction models of 1/3/5-year OS and validated the biological roles of *ADM* and *POLR1D* genes in OSCC cell lines through gene knockdown experiments. Additionally, gene set enrichment analysis was performed to identify signaling pathways enriched by the prognostic features associated with 6-GPS. In conclusion, this study focused on OSCC CSCs and developed and validated a new prognostic feature associated with cancer stem cell marker genes, which can be used to guide survival risk stratification and improve the management of OSCC patients.

## 2. Results

### 2.1. Identification of CSCMGs Expression Profiles

Data quality control (QC) and visualization were carried out by the Seurat package, and the gene numbers, cell numbers, and mitochondrial contents were calculated (Appendix A). After the QC and the removal of batch effects, multiple principal components (PC) populations with large differences that can be used as anchor points were obtained (shown as PC1 and PC2), and the results showed that the batch removal effect was excellent (Appendix A). Next, HVGs were calculated and visualized, and the top 10 HVGs were marked (Appendix A). Then, the top 2000 HVGs were used for further PC analysis. The ElbowPlot function was used to evaluate the PCs, and the results were visualized as the scree plot (Appendix A).

Principal Component Analysis (PCA) and scree plot analysis were performed to investigate the data structure, followed by Uniform Manifold Approximation and Projection (UMAP) dimensionality reduction analysis, which enabled the identification of 19 distinct cell clusters (Figure 1A). Subsequently, the reference dataset from the human main cell map was employed to annotate the cell types present in each cluster. Among the 14 annotated cell clusters, a total of 15,731 unique gene expressions were observed, as documented in Appendix A. Notably, Cluster 1 encompassed cells classified as CSCs (Figure 1B), consisting of 1344 genes with expression profiles specifically associated with CSCs, as indexed in Appendix A. To further ascertain this distinction, the expression levels of cancer stem cell markers *TACSTD2* [24] and *KRT19* [25] were assessed across all cell clusters. Interestingly, Cluster 1 exhibited significantly higher expression of both *TACSTD2* (Figure 1C,E) and *KRT19* (Figure 1D,F) when compared to other cell clusters.

### 2.2. Establishment of the 6-GPS Based on CSCMGs

Utilizing a predetermined cutoff value, we subjected 124 CSCs-related genes to univariate Cox regression analysis to establish a prognostic feature among the 1344 identified genes (Appendix A). From this analysis, 19 CSCMGs were found to be significantly associated with the prognosis of OSCC, as determined by single-variable Cox regression analysis (Figure 2).

Employing the least absolute shrinkage and selection operator (LASSO) Cox regression analysis, we further identified the six most predictive genes (*ADM*, *PTGR1*, *RPL35A*, *PGK1*, *POLR1D*, and *P4HA1*) to construct a prognostic risk scoring model (Figure 3A,B). The resulting model, known as the 6-GPS, was validated as an independent prognostic factor with remarkable statistical significance (*p* < 0.01). Subsequently, for each sample, the risk score was calculated using the 6-GPS and the corresponding regression coefficient. This calculation was performed with the following equation: Risk-based score = Sum of each gene (Regression coefficient × Gene expression). The optimal risk score cutoff value was determined based on the median risk score using the ‘SurvMiner’ package within the training set. Based on their respective risk scores, patients were classified into high-risk and low-risk groups, with the dividing line set at the median risk score (Figure 3C). The distribution of survival status was visually represented in Figure 3D, respectively, clearly demonstrating a higher incidence of mortality within the high-risk group. Figure 3E provides detailed expression profiles of the six genes comprising the 6-GPS. Kaplan-Meier (K-M) analysis revealed a significantly inferior OS for patients with high-risk scores as compared to those with low-risk scores (Figure 3F). To evaluate the predictive performance of the risk model, we determined the area under the time-dependent receiver operating characteristic (TimeROC) curve (AUC) at different time points (1, 3, and 5 years), which yielded AUC values of 0.696, 0.664, and 0.636, respectively (Figure 3G).

### 2.3. Construction and Assessment of Nomogram

A widely used prognostic visualization tool in oncology, the nomogram, was employed to estimate patient survival by incorporating an index score. Based on our predictive model, patients who are over 60 years old, female, in clinical stages Ⅲ to Ⅳ, and have high expression of *ADM*/*PTGR1*/*RPL35A*/*PGK1*/*POLR1D*/*P4HA1* are more likely to be in a high-risk state (Figure 4A). These predictive factors can serve as important criteria for evaluating patient risk and devising personalized treatment strategies. The calibration curve serves as evidence of the effective performance of the prognostic gene traits in predicting survival outcomes, with the nomogram accurately estimating 1-, 3- and 5-year OS (Figure 4B). The Y-axis represents the actual survival probability, while the X-axis represents the estimated survival probability generated by the nomogram. The calibration curve, characterized by a 45-degree diagonal dotted line, indicates perfect agreement between the observed probability and the actual probability.

### 2.4. Validation of the Prognostic Signature

The ICGC cohort was employed as the validation cohort for 6-GPS, and the patients were split into low-risk and high-risk groups based on the training cohort’s median risk score. Figure 5A–C display the 6-GPS risk score, survival status, and heatmap for the validation cohort. The survival curve demonstrated that people in the high-risk category had a worse prognosis than those in the low-risk category, which was consistent with the findings of the training cohort (Figure 5D). Additionally, the 1-, 2-year OS AUC prediction values were 1.000 and 0.953, respectively (Figure 5E). Overall, 6-GPS has demonstrated worldwide applicability in its performance as a classifier.

### 2.5. Expression Levels of 6-GPS

The mRNA expression of the 6-GPS was subjected to analysis using the Cancer Genome Atlas (TCGA) dataset. In comparison to the established normal standard, the OSCC samples exhibited significantly lower expression of *PTGR1*, while demonstrating higher expression of *ADM*, *RPL35A*, *PGK1*, *POLR1D*, and *P4HA1* (*p* < 0.05, Figure 6A,B). Correspondingly, the HPA website conducted evaluations of the protein abundance of the 6-GPS. Immunohistochemistry results revealed that OSCC samples exhibited notably higher protein levels of PTGR1, PGK1, POLR1D, and P4HA1 compared to normal skin and mucosa tissues (Figure 6C–J). Importantly, patients diagnosed with OSCC displayed a distinct survival disadvantage in the high expression group of the 6-GPS, as demonstrated by lower OS rates in comparison to the low expression group (Figure 6K–P). Collectively, these comprehensive findings provide compelling evidence supporting a robust association between the 6-GPS and the prognosis of OSCC patients.

### 2.6. Analysis of the Functional Characteristics of 6-GPS

To gain deeper insights into the underlying biological pathways associated with the 6-GPS, we conducted further investigation by identifying genes correlated with the 6-GPS and constructing a protein–protein interaction (PPI) network analysis map, depicted in Figure 7A. Subsequently, the interactions among these genes were explored using GeneMANIA (http://genemania.org). The resultant analysis revealed that the 6-GPS interaction network was significantly enriched in biological processes related to response to reduced oxygen levels, hypoxia, cellular response to oxygen levels, as well as protein hydroxylation, protein targeting to the endoplasmic reticulum (ER), protein localization to the ER, and protein targeting to the cell membrane (Figure 7B). Moreover, functional enrichment analyses using Gene Ontology (GO) and Kyoto Encyclopedia of Genes and Genomes (KEGG) were performed. The results generated three functional categories encompassing biological processes (BP), cellular components (CC), and molecular functions (MF), which are detailed in Appendix A. Notably, the enrichment analysis unveiled that the expression of the 6-GPS was closely associated with various functional annotations such as the ribosome, hypoxia-inducible factor 1 (HIF-1) signaling pathway, and ribosome structural component. Furthermore, examples of enriched terms include protein targeting to the cell membrane, nuclear-transcribed mRNA catabolism, cellular response to oxygen levels, cellular response to hypoxia, and protein localization to the ER (Figure 7C,D). These findings collectively suggest that the 6-GPS is significantly implicated in biological pathways relevant to cancer progression.

### 2.7. Inhibition of OSCC Cell Proliferation, Migration, and Invasion upon ADM and POLR1D Genes Knockdown

Previous studies have elucidated the roles of other genes within the 6-GPS in OSCC. However, the biological functions of *ADM* and *POLR1D* remain to be validated. To further investigate the biological effects of *ADM* and *POLR1D* in CAL-27 and SAS cell lines, we utilized siRNA knockdown technology. Real-time quantitative polymerase chain reaction (RT-qPCR) results showed that siRNA3-*ADM* and siRNA3-*POLR1D* exhibited high knockdown efficiency in CAL-27 and SAS cells (Figure 8A,B); thus, these two siRNAs were selected for subsequent experiments. CCK-8 assays revealed that interfering with targeted gene expression significantly inhibited the proliferation of OSCC cells (Figure 8C,D). The Transwell migration and invasion assay confirmed that *ADM* or *POLR1D* knockdown inhibited the migration (Figure 8E,F) and invasion (Figure 8G,H). Subsequent RT-qPCR analysis unveiled a reduction in *MMP2* mRNA expression subsequent to the knockdown of *ADM* and *POLR1D* (Figure 8I). Furthermore, flow cytometry analysis showed that knockdown of *ADM* or *POLR1D* induced G1 phase cell-cycle arrest in CAL-27 and SAS cells (Figure 8J). These findings corroborate the bioinformatics analysis, indicating that *ADM* and *POLR1D* may promote the proliferation and tumorigenicity of OSCC cells.

### 2.8. The Mechanistic Role of ADM and POLR1D in OSCC Cells

The cell surface antigen CD133 is considered to be a cancer stem cell biomarker. Flow cytometry analysis revealed a significant decrease in CD133-positive cells in CAL-27 and SAS cell lines upon knockdown of *ADM* and *POLR1D* genes compared to the control group (Figure 9A,B). RT-qPCR results demonstrated a correlation between the *ADM* and *POLR1D* genes, as the knockdown of *ADM* or *POLR1D* resulted in the downregulation of the other gene. Additionally, the study investigated the related genes in the PPI network. Interestingly, the knockdown of *ADM* led to compensatory upregulation of its receptor gene, *RAMP2*. On the other hand, the expression of *POLR1F* and *POLR3B* decreased with the knockdown of *POLR1D* (Figure 9C,D). The analysis of gene enrichment demonstrated the connection between 6-GPS and the HIF-1 signaling pathway, cellular response to hypoxia, and other relevant pathways. Spearman correlation analysis revealed a correlation between the *ADM* (Figure 9E) and *POLR1D* (Figure 9F) genes and the cellular response to hypoxia. To investigate the downstream effects of *ADM* and *POLR1D* knockdown in SAS cells, we performed Western blot analyses to assess the expression levels of key genes. Our findings revealed a decrease in the levels of JAK1, phosphorylated-JAK1, and HIF-1α expression following the knockdown of the target genes. Knockdown of *ADM* resulted in reduced expression of Cyclin D1, a marker associated with the transition from G1 to S phase. Furthermore, the knockdown of *POLR1D* was accompanied by decreased CDK2 expression. Notably, we also observed a significant reduction in the protein levels of CD133 and ALDH1A1, markers indicative of cancer stem cells (Figure 9G,H). These findings suggest that the *ADM* and *POLR1D* genes may play a role in the cellular response to hypoxia and potentially regulate critical pathways associated with the properties of cancer stem cells in OSCC cells.

## 3. Discussion

A growing body of research demonstrates that CSCs are crucial to the formation and development of OSCC [26]. Patients with OSCC may live longer if cancer stem cells are specifically targeted [27,28]. The advancement of bioinformatics enables comprehensive investigation into the molecular characteristics of OSCC and the identification of potential prognostic biomarkers. For instance, Wang et al. [4] obtained gene expression profiles from the GEO database and identified the prognostic signature of six genes in OSCC through bioinformatics analysis and RT-qPCR validation. Wu et al. [29] constructed an OSCC-related risk model based on cancer-associated fibroblasts by scRNA-seq analysis and explored the potential correlation between OSCC and periodontal disease. However, current research lacks the investigation of potential mechanisms underlying the occurrence and progression of OSCC through the establishment of prognostic signatures based on CSCMGs. Additionally, the limited sample size of individual GEO datasets may result in biases in differential gene studies without yielding biological significance. In contrast to the analysis of a single dataset, the present study used the GEO database to build a novel predictive signature for OSCC patients based on CSCMGs, which was then thoroughly verified in the ICGC cohort. Moreover, we validated the reliability of these prognostic signatures through experimental validation.

LASSO-Cox regression analyses were employed to establish a prognostic signature, leading to the discovery of six genes (*ADM*, *POLR1D*, *PTGR1*, *RPL35A*, *PGK1*, and *P4HA1*) as constituents of the prognostic signature. A nomogram incorporating clinical characteristics was constructed based on the ability of these genes to distinguish between OSCC patients at different risk levels. The prognosis of high-risk patients was found to be worse compared to that of low-risk individuals. The nomogram, calibration plot, and ROC curve data collectively provided strong evidence supporting the prognostic accuracy of the 6-GPS. Furthermore, in vitro experiments validated the inhibitory effects on cell proliferation, migration, and invasion after the knockdown of *ADM* and *POLR1D* genes, with decreased expression levels of CSCs markers, such as the cell surface antigen CD133 and ALDH1A1.

*ADM*, which was originally identified as a hypotensive and vasodilatory peptide belonging to the calcitonin gene-related peptide family, has been observed to exert influence on vascular development, cell proliferation, migration, apoptosis, and differentiation [30]. ADM and its receptor modifying proteins, RAMPs, upregulation has been detected in diverse human malignancies relative to healthy tissue [31]. Previous studies have revealed that ADM-RAMP2 exerts a stimulatory effect on tumor angiogenesis, leading to the suppression of tumor cell adhesion and invasion. The loss of RAMP2 promotes the formation of pre-metastatic niches in distant organs by disrupting vascular structure and inducing inflammation [32,33]. In the context of acute myeloid leukemia, the expression of *ADM* is linked to stem cell phenotypes, inflammatory attributes, and genes associated with immune suppression [34]. Similarly, the presence of *ADM* in solid tumors has been found to enhance the probability of lymph node metastasis in ovarian cancer [35]. Furthermore, *ADM* has been established as a dependable prognostic marker for long-term outcomes in individuals with nasopharyngeal carcinoma [36]. Our study demonstrates a positive correlation between elevated *ADM* expression and poor prognosis in OSCC patients. In vitro experiments further substantiate these findings, as *ADM* knockout inhibits proliferation, migration, and invasion of OSCC cells, reduces mRNA levels of *MMP2*, and induces compensatory upregulation of *RAMP2*, therefore suppressing tumor cell migration and invasion. Furthermore, the depletion of *ADM* results in decreased expression of Cyclin D1 protein, leading to cell-cycle arrest at the G1-S phase in OSCC cell lines.

The *POLR1D* gene is responsible for encoding a subunit of RNA polymerase I and RNA polymerase III. It is located on chromosome 13q [37]. A deficiency in its functionality leads to impaired ribosome biogenesis, which in turn affects crucial cellular processes such as cell death, proliferation, and differentiation [38,39]. In the context of colorectal cancer research, *POLR1D* has been found to promote the G1-S cell-cycle transition by activating the Wnt-β-Catenin pathway and suppressing the p53 signaling pathway [40]. Additionally, *POLR1D* has been observed to influence the expression of *VEGF-α* and *EREG*, resulting in acquired resistance to bevacizumab in colorectal cancer [41]. Our study indicates that high expression of *POLR1D* is an independent adverse prognostic factor in OSCC. Knockdown of *POLR1D* reduces the expression levels of CDK2 and Cyclin D1, leading to the induction of G1-S phase cell-cycle arrest and significant inhibition of the carcinogenic behavior of OSCC cells.

Since *PTGR1* is overexpressed in several cancer cell lines, including those from stomach [42], liver [43], lung [44], and prostate [45] cancers, *PTGR1* may cause cancer. Conversely, diminished expression of *PTGR1* is correlated with unfavorable prognosis in ovarian cancer, endometrial cancer, and renal clear cell carcinoma [46]. In the investigation of breast cancer stem cells, *PTGR1* has been recognized as a prospective prognostic indicator associated with the condition of breast CSCs and the progression of cancers [47]. Moreover, the elevation of anti-PTGR1 autoantibody in saliva is considered a biomarker for early-stage OSCC [48]. The *RPL35A* gene is situated on the chromosomal band 3q29-qter and plays a crucial role in facilitating protein synthesis, therefore serving as a biomarker for cancer angiogenesis [49,50]. Recent investigations have revealed that *RPL35A* potentially facilitates the advancement of gastric cancer by means of the p38/JNK signaling pathway [51]. Our observations indicate that this phenomenon is also evident in OSCC tissue, particularly characterized by elevated levels of RPL35A in cancer tissues, which correlates with an unfavorable prognosis. As a glycolytic enzyme, *PGK1* plays a crucial role not only in tumor energy metabolism but also in regulating and expressing multiple cancer proteins [52]. In the treatment of liver cancer, breast cancer, colorectal cancer, and prostate cancer, *PGK1* is considered a critical target. Previous studies have demonstrated that the *PGK1* and *P4HA1* genes serve as prognostic features of OSCC as hypoxia-related genes [53]. In the hypoxic environment of OSCC, the activated AKT signaling pathway increases the expression of *PGK1*, promoting glycolysis, enhancing stem-like properties, and epithelial-mesenchymal transition (EMT) [54,55,56,57]. The expression of *P4HA1* is markedly elevated in OSCC tissues, and heightened *P4HA1* expression is linked to a more unfavorable prognosis [58]. Furthermore, a robust correlation exists between *P4HA1* mRNA levels and various EMT and stem cell markers [59]. These investigations substantiate the pivotal involvement of the 6-GPS in the development of OSCC and propose that the genes identified in CSCMGs might serve as promising targets for laboratory-based experimentation.

Results of the 6-GPS interaction network analysis primarily encompassed responses to decreased oxygen levels, hypoxic reactions, and cellular responses to oxygen levels. Spearman correlation analysis revealed a correlation between *ADM* and *POLR1D* with cellular response to hypoxia. Knockdown of these two genes resulted in reduced levels of JAK1/p-JAK1/HIF-1α proteins in CAL-27 and SAS cells, suggesting that *ADM* and *POLR1D* may impact the hypoxic microenvironment in OSCC via the JAK/HIF-1 pathway, therefore influencing its development. Through GO and KEGG analysis, a total of 11 enrichment terms and 1 signaling pathway were identified. These terms and pathways play crucial roles in various stages of OSCC development, particularly involving ribosome biogenesis, mRNA degradation in the nucleus, protein targeting to membranes, cellular response to oxygen levels, cellular response to hypoxia, and the HIF-1 signaling pathway.

Despite positive findings, it is important to note that our study has several limitations. First, a sizable OSCC research institution needs to conduct a prospective study on the accuracy and stability of our prognostic signature. Second, based on the insufficient experimental results mentioned above, it cannot be concluded that *ADM* and *POLR1D* can serve as cancer stem cell markers in OSCC. Future studies should incorporate additional experiments to validate the role of these two genes in CSCs. In summary, the relationship between the expression of the 6-GPS and the prognosis for OSCC should be further investigated in future studies.

Taken together, a remarkable performance in predicting the prognosis of OSCC patients has been found and proven for the six-gene prognostic signature based on CSC marker genes. It is a prognostic biomarker that may be applied in clinical settings to provide personalized predictions. Furthermore, the inhibitory effects of knocking down *ADM* and *POLR1D* expression on cancer-related biological behaviors in CAL-27 and SAS cells were experimentally validated. Thus, *ADM* and *POLR1D* might become a promising therapeutic target for OSCC.

## 4. Materials and Methods

### 4.1. Data Download and Preprocessing

To identify the CSCMGs of OSCC, the scRNA-seq data set (GSE172577) of OSCC [60] was obtained and preprocessed from the GEO database (https://www.ncbi.nlm.nih.gov/geo) (accessed on 17 October 2022). TCGA dataset, along with clinical information for 40 OSCC patients, was collected from the ICGC database (https://dcc.icgc.org/releases/current/Projects) (accessed on 3 November 2022) for CSCMGs localization and the derivation of predictive characteristics. Relevant immunohistochemical information of OSCC patients was downloaded from the Human Protein Atlas (HPA) database (https://www.proteinatlas.org) (accessed on 21 January 2023) for further validation. Initially, the R package Seurat [version 3.0.2] was utilized to standardize the scRNA-seq data. The normalization process involved using the “Normalize Data” function with the normalization technique set to “Log Normalize”. Subsequently, the transformed data were converted into a Seurat object. To ensure the retention of high-quality data, three filtering criteria were applied to the raw scRNA-seq data matrix using the R package Seurat. Genes detected in fewer than 3 cells, cells expressing fewer than 200 genes or more than 7000 genes, and cells with a high mitochondrial proportion (>10%) were removed. The “Find Variable Features” function and the “Harmony” method were employed to identify the top 2000 highly variable genes (HVGs). It is important to note that this study utilized publicly accessible datasets from the initial research, which had received ethical approval.

### 4.2. Identification of CSCs Marker Genes by scRNA-seq Analysis

PCA utilizing the “Run PCA” function implemented in the R package Seurat. To identify the most significant PC population for gene population clustering, the relevance of PCs was determined through Jack Straw analysis, enabling the identification of potential “magnets” for PC populations. Subsequently, the first 17 PCs were selected for visualization using the UMAP technique, which served as a reduction technique for dimensionality clustering. Cell cluster analysis was performed using the “Find Clusters” function with a resolution parameter of 0.5. Differentially Expressed Genes within each cluster were identified using the “Find All Markers” function, and the Wilcoxon–Mann–Whitney test was applied. Marker genes for each cluster were determined based on an adjusted *p*-value *<* 0.01 and *|log*_2_*FC|* > 1 [61,62,63,64]. Annotation of all clusters was carried out manually, utilizing known cell-type marker genes.

### 4.3. Construction of 6-GPS Based on CSCMGs

To enhance the comparability between TCGA samples and microarrays, the RNA-seq data obtained from the TCGA Fragments Per Kilobase Per Million format was transformed into transcripts per million reads (TPM) values and subjected to log2-transformation. Univariate Cox regression analysis was employed to evaluate the predictive significance of CSCMGs for OS in TCGA OSCC patients, with a prognostic gene defined as having a *p*-value *<* 0.05. To mitigate overfitting, prognostic gene analysis was performed using the LASSO-Cox proportional hazards regression, utilizing the “survival” package [version 3.2-10] and the “glmnet” package [version 4.1-2]. LASSO is a widely utilized regression technique for high-dimensional variables in Cox proportional hazard regression models for survival analysis. To select the optimal model, 10-fold cross-validation was performed using the “cv.glmnet” function, with the tuning parameter LASSO selected based on 1-SE (standard error). Genes with beta coefficients greater than zero were included in the Gene Prognostic Signature, which was constructed based on the linear combination of mRNA expression and the associated hazard coefficient identified by LASSO-Cox regression analysis. The prognostic value of the Gene Prognostic Signature was established using a dichotomized approach, with patients being categorized into low-risk and high-risk groups based on the median cutoff value. The impact of the 6-GPS on OSCC patient prognosis in the TCGA dataset was assessed using K-M survival curves. The AUC was calculated using the “TimeROC” package [version 0.4], and the prognostic capability of the 6-GPS was evaluated by constructing risk factor groups using the “ggplot2” package [version 3.3.3]. The “Survival” package [version 3.2-10] was used for the statistical analysis of the survival data. The statistical significance of the difference was determined using the log-rank test implemented in the “SurvMiner” package [version 0.4.9].

### 4.4. Development of a Prognostic Nomogram and Assessment of Its Predictive Performance

To estimate the survival rates at 1, 3, and 5 years, a nomogram was constructed using the “RMS” package [version 6.2-0]. Additionally, calibration curves were generated to assess the agreement between the predicted survival probabilities from the nomogram and the actual observed survival outcomes.

### 4.5. Validation of the Prognostic Signature

The RNA-seq data and associated clinical information of 40 cases of oral malignancies were obtained from the ICGC database for the purpose of externally validating and verifying the wide applicability of the 6-GPS. The risk score for each individual in the validation cohort was determined by applying the risk score equation derived from the training cohort. Consequently, the individuals in the validation cohort were classified into low-risk and high-risk groups based on the risk threshold established in the training cohort. Utilizing the aforementioned risk stratification, K-M survival curves and TimeROC curves were promptly generated, as previously described.

### 4.6. Pathway and Function Enrichment Analysis

To gain a comprehensive understanding of the functional characteristics of the 6-GPS and its potential associated signaling pathways, gene enrichment analysis using the KEGG and GO databases was conducted employing the “clusterprofiler” package [version 3.14.3]. Additionally, the “org.Hs.eg.db” package [version 3.10.0] was employed for efficient conversion and mapping of gene identifiers, facilitating a more informative interpretation of the 6-GPS [65].

### 4.7. Correlation between ADM/POLR1D and the Cellular Response to Hypoxia

To examine the influence of *ADM* and *POLR1D* on the cellular response to hypoxia, we assembled a collection of genes associated with the hypoxia pathway. The R software package version 4.0.3, in conjunction with the GSVA package, was employed for pathway analysis using the method = ‘ssgsea’. Subsequently, Spearman correlation analysis was conducted to evaluate the association between gene expression and pathway scores.

### 4.8. Cell Culture and Transfection

The OSCC cell lines CAL-27 (Servicebio, Wuhan, China) and SAS (Cellcook, Guangzhou, China) were cultivated in Dulbecco Modified Eagle Medium F12 (DMEM/F12) (Servicebio, Wuhan, China) supplemented with 10% fetal bovine serum (FBS) (ABW, Shanghai, China) and 100 U/mL penicillin-streptomycin-gentamicin solution (Solarbio, Wuhan, China). Culturing was performed in a CO_2_-controlled incubator at 37 °C. A density of 1 × 10^6^ cells was seeded in T-25 cell culture flasks. For transfection, siRNAs targeting *ADM* (siRNA1-*ADM*, siRNA2-*ADM*, siRNA3-*ADM*), siRNAs targeting *POLR1D* (siRNA1-*POLR1D*, siRNA2-*POLR1D*, siRNA3-*POLR1D*), and a negative control (siRNA-NC) (GenePharma, Shanghai, China) were transfected into cells using the Lipofectamine™ 2000 transfection kit (Life Technologies, Carlsbad, CA, USA). After 24 h, the expression levels of *ADM* and *POLR1D* were assessed via RT-qPCR.

### 4.9. RT-qPCR

The SPARKeasy Cell RNA Kit (Sparkjade, Qingdao, China) was utilized for the extraction of total RNA from the transfected cells. The same amount of total RNAs was reversed to cDNA according to the Reverse Transcription Kit manufacturer’s protocol (AG11705, Accurate Biology). Afterwards, RT-qPCR was conducted utilizing the QIAGEN Rotor-GeneQ (QIAGEN, Dusseldorf, Germany) with SYBR Green Pro. Tag. HS PremixⅡ (AG11702, Accurate Biology, Changsha, China). The thermocycling conditions were as follows: pre-denaturation at 95 °C for 30 s, followed by 40 cycles of denaturation at 95 °C for 5 s, and extension at 60 °C for 30 s. All experiments were repeated three times independently, and each sample underwent a melting curve analysis to verify the specificity of amplification. Table 1 presents the primer sequences. The expression of the target genes was normalized to GAPDH, and fold changes were calculated in the manner of the 2^−ΔΔCT^.

### 4.10. Cell Proliferation Assay

CAL-27 and SAS cells were seeded in 96-well plates at a density of 5000 cells per well. At designated time points (0, 24, 48, and 72 h), 10μL of CCK-8 reagent (Apexbio, Houston, TX, USA) was added to each well and incubated for 2 h. The optical density (OD) readings at a wavelength of 450 nm were obtained using a microplate reader (Infinite M200 Pro, Tecan Group, Zurich, Switzerland).

### 4.11. Cellular Migration and Invasion Assays

Invasion of cells was assessed using Transwell inserts coated with Matrigel, whereas 8 μm Transwell inserts were employed to measure cellular migration. CAL-27, as well as SAS cells following transfection, were seeded onto the upper chambers (1 × 10^5^ cells/well). DMEM plus 10% FBS was added into the lower chambers. Following 36 h, the cells from the upper surface of the membrane were wiped off using a cotton swab. The invaded cells were stained with 0.4% crystal violet. Five fields were counted under a microscope (Olympus, Tokyo, Japan). The quantification of the results was performed using ImageJ [ImageJv1.8.0.322], followed by data analysis using GraphPad Prism [version 8.0.2].

### 4.12. Flow Cytometry Assay

Following transfection, CAL-27 and SAS cells were trypsinized using a trypsin solution without ethylenediaminetetraacetic acid. The cells were then collected by centrifugation at 1000× *g* for 5 min, washed with ice-cold PBS, and fixed with 70% cold ethanol for storage at 4 °C for 24 h. After another round of centrifugation and washing with cold PBS, the cells were treated with 25 μL Propidium and 10 μL RNase A (Beyotime, Shanghai, China) at 37 °C in the dark for 30 min. For CD133 detection, cells were obtained and stained with anti-CD133 (17–1338-42, Thermo Fisher Scientific, Waltham, MA, USA) at 4 °C in the dark for 30 min. Cell-cycle analysis was conducted using NovoExpress [Version 1.6.2, Agilent, Santa Clara, CA, USA], and the CD133 was analyzed using CytExpert software [Version 2.4.0.28, Beckman Coulter, Brea, CA, USA].

### 4.13. Western Blot Analysis

SAS cell was washed with PBS and lysed in RIPA buffer with PMSF (Solarbio, Beijing, China) on ice. Cell lysates were centrifuged (12,000 rpm) at 4 °C for 10 min and then quantified using the BCA Protein Assay Kit (Cwbio, Taizhou, China). The lysate was denatured with sodium dodecyl sulfate–polyacrylamide gel electrophoresis (SDS–PAGE) sample loading buffer (Solarbio, Beijing, China), followed by SDS–PAGE and electrotransfer to polyvinylidene difluoride membranes (Cytiva, Washington, DC, USA). The membranes were incubated overnight at 4 °C with anti-β-actin (Proteintech Group, 66009-1-Ig, 1:5000, Wuhan, China), anti-JAK1 (Proteintech Group, 66466-1-Ig, 1:4000, Wuhan, China), anti-phospho-JAK1 (Servicebio, GB115604-100, 1:300, Wuhan, China), anti-HIF-1α (Proteintech Group, 66730-1-Ig, 1:3000, Wuhan, China), anti-CDK2 (Proteintech Group, 60312-1-Ig, 1:2000, Wuhan, China), anti-Cyclin D1 (Proteintech Group, 60186-1-Ig, 1:1000, Wuhan, China), anti-CD133(Proteintech Group, 66666-1-Ig, 1:3000, Wuhan, China), and anti-ALDH1A1 (Proteintech Group, 60171-1-Ig, 1:5000, Wuhan, China) at the appropriate dilution and then incubated with the secondary antibody at room temperature for 2 h. The bands were visualized by employing an ECL detection reagent (Yeasen, Shanghai, China) and independently quantified twice using ImageJ [ImageJv1.8.0.322] and ChemiScope Analysis [Version 2.1.6.0] software. Subsequently, the obtained data were normalized to β-actin.

### 4.14. Statistical Analysis

Statistical analyses and data visualization were performed using R software [version 3.6.3, R Core Team, Vienna, Austria] and Adobe Illustrator [Version 25.0.0.60, Adobe Inc., Mountain View, CA, USA], respectively. Cox regression analysis was employed to explore the association between clinical variables and gene expression levels. Descriptive statistics were presented as mean ± standard deviation. Statistical differences between groups were assessed using two-tailed *t*-tests or analysis of variance. *p*-value *<* 0.05 was the cutoff criterion.

## Figures and Tables

**Figure 1 ijms-25-01031-f001:**
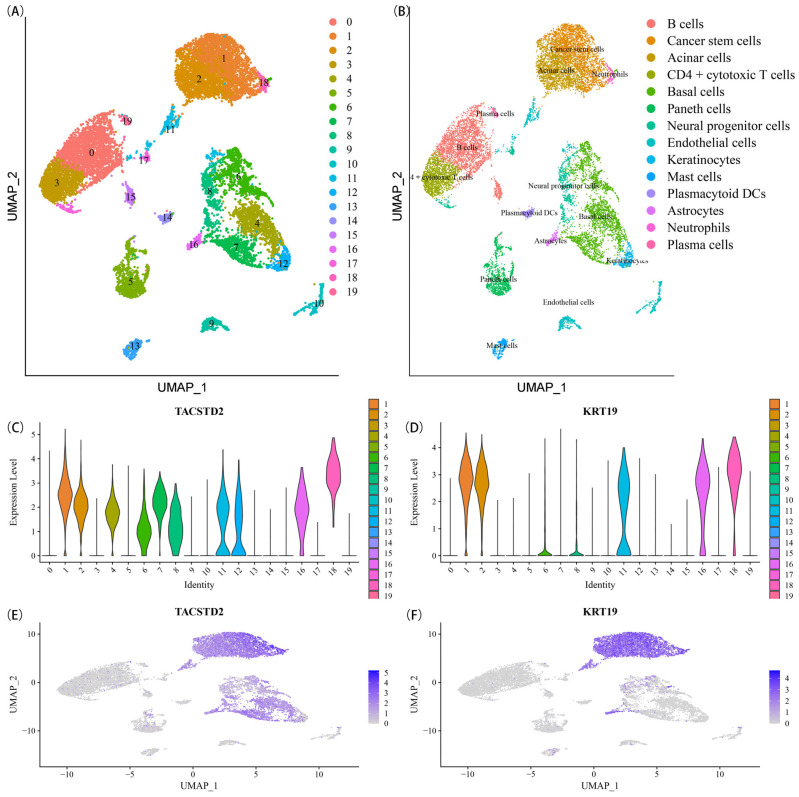
Cancer stem cell clusters were discovered using scRNA-seq analysis. (**A**) The UMAP map is colored by various cell clusters; (**B**) Cell types recognized by marker genes; (**C**) The expression level of *TACSTD2* in each cell cluster; (**D**) The expression level of *KRT19* in each cell cluster; (**E**) *TACSTD2* expression details; (**F**) *KRT19* expression details.

**Figure 2 ijms-25-01031-f002:**
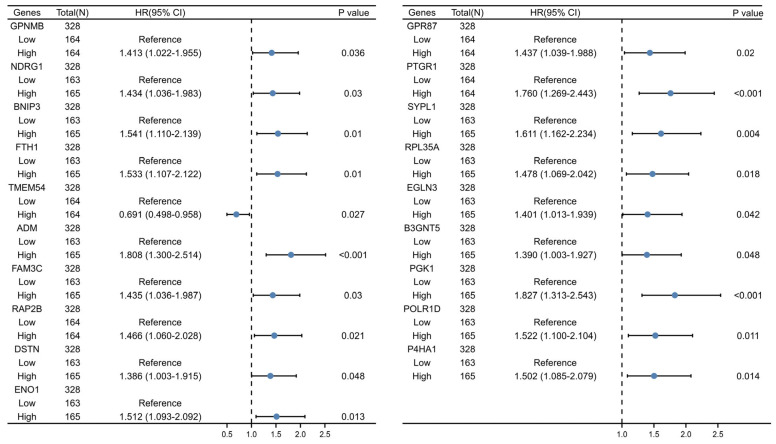
Univariate Cox regression analysis was used to screen the genes related to the prognosis of OSCC (*p* < 0.05 was statistically significant).

**Figure 3 ijms-25-01031-f003:**
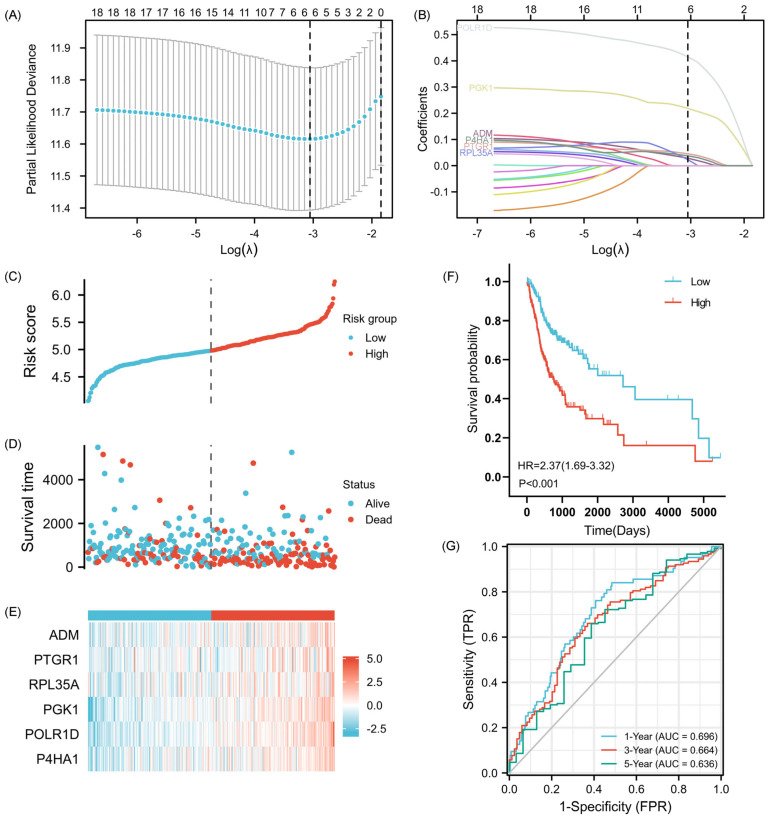
Establishment of a prognosis-associated signature for OSCC. (**A**) 6-GPS-based LASSO cross-validation plot. (**B**) LASSO coefficient of 6-GPS in OSCC. (**C**) Visualization of risk scores of OSCC patients. (**D**) Visualization of dead and alive OSCC patients with high- and low-risk scores. (**E**) Heatmap of the expression of the 6-GPS in specimens with high-risk and low-risk scores. (**F**) Survival analyses of OSCC subjects with high- and low-risk scores with K-M curves. (**G**) Estimation of the predictive ability of the nomogram in OSCC prognosis using the ROC curves.

**Figure 4 ijms-25-01031-f004:**
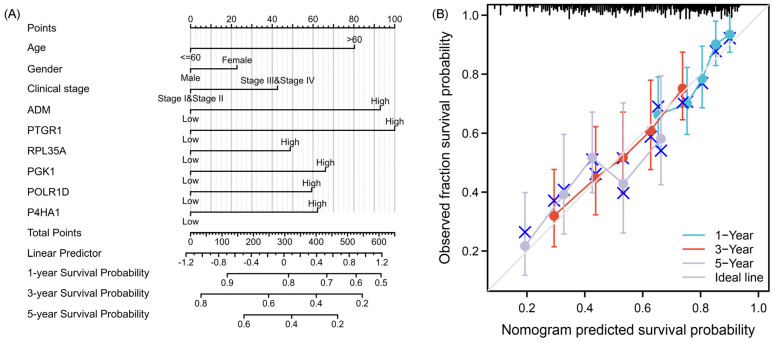
Estimated prognostic accuracy of 6-GPS in patients with OSCC. (**A**) Nomogram shows that the age, gender, clinical stage, and risk score were associated with 1-, 3-, and 5-year OS. (**B**) Calibration plots for showing the deviation between model-estimated and observed 1-, 3-, and 5-year survival.

**Figure 5 ijms-25-01031-f005:**
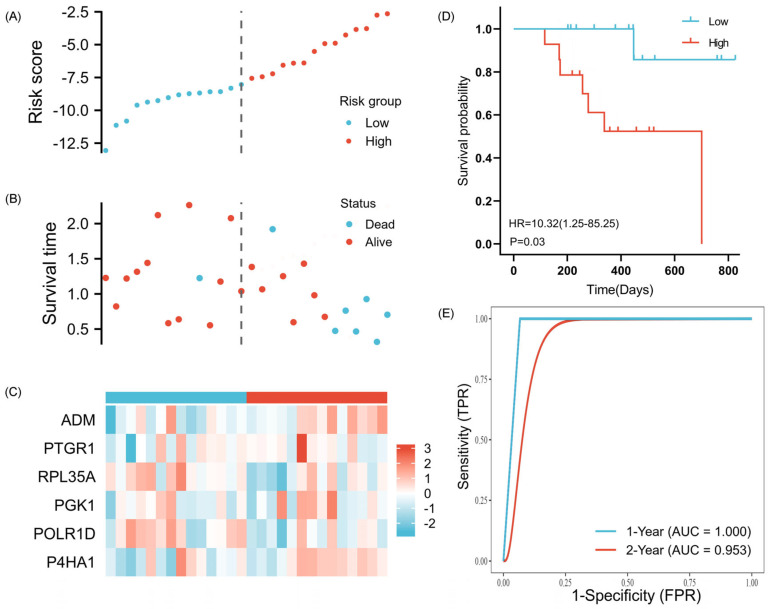
Risk score analysis of 6-GPS in the ICGC. (**A**–**D**) The risk score, survival status of OSCC patients, heatmap of the 6-GPS expression, and survival curves between low and high-risk groups are shown. (**E**) Time-independent ROC analysis of risk score for predicting the OS in the ICGC cohort.

**Figure 6 ijms-25-01031-f006:**
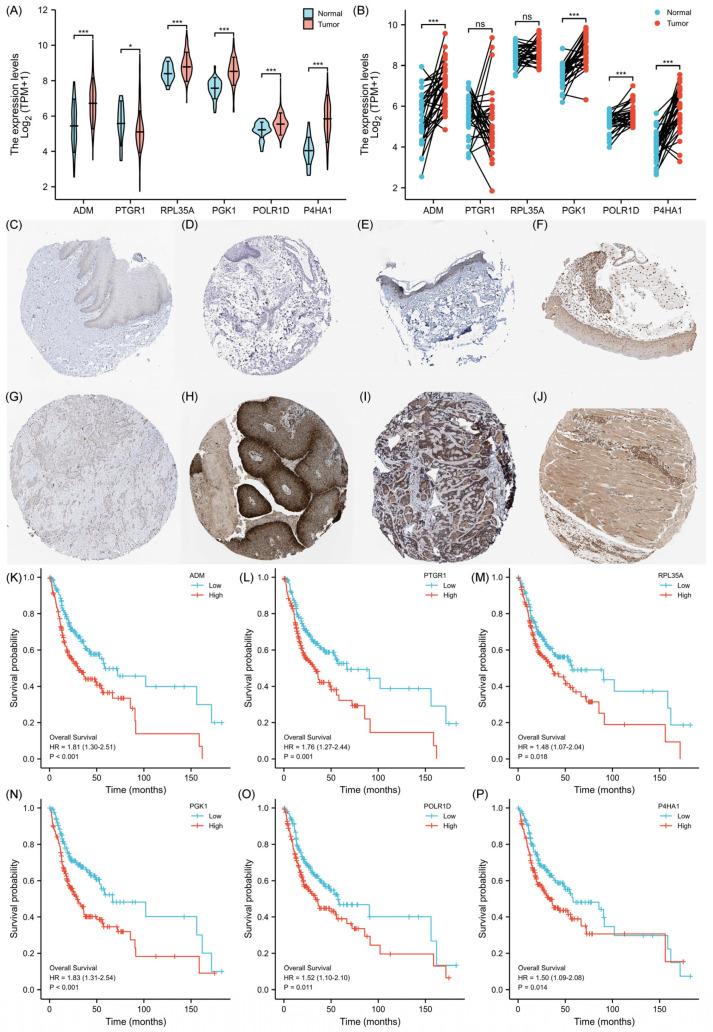
Expression levels of 6-GPS (**A**) Comparing the 6-GPS expression levels in all OSCC and normal epithelium tissues. (**B**) Comparison of 6-GPS expression levels between cancer and adjacent normal tissues from the same patient. (**C**) Negative expression of PTGR1 was observed in normal oral mucosa (HPA036724). (**D**) Negative expression of PGK1 was observed in normal oral mucosa (HPA073644). (**E**) Medium expression of POLR1D was observed in normal skin (25–75%, HPA039337). (**F**) Medium expression of P4HA1 was observed in normal oral mucosa (>75%, HPA026593). (**G**) Low expression of PTGR1 was observed in OSCC (HPA036724). (**H**) High expression of PGK1 was observed in OSCC (>75%, CAB010065). (**I**) Medium expression of POLR1D was observed in OSCC (>75%, HPA039337). (**J**) High expression of P4HA1 was observed in OSCC (>75%, HPA026593). (**K**–**P**) K-M survival curves for the 6-GPS. (*: *p* < 0.05; ***: *p* < 0.001; ns: No significance).

**Figure 7 ijms-25-01031-f007:**
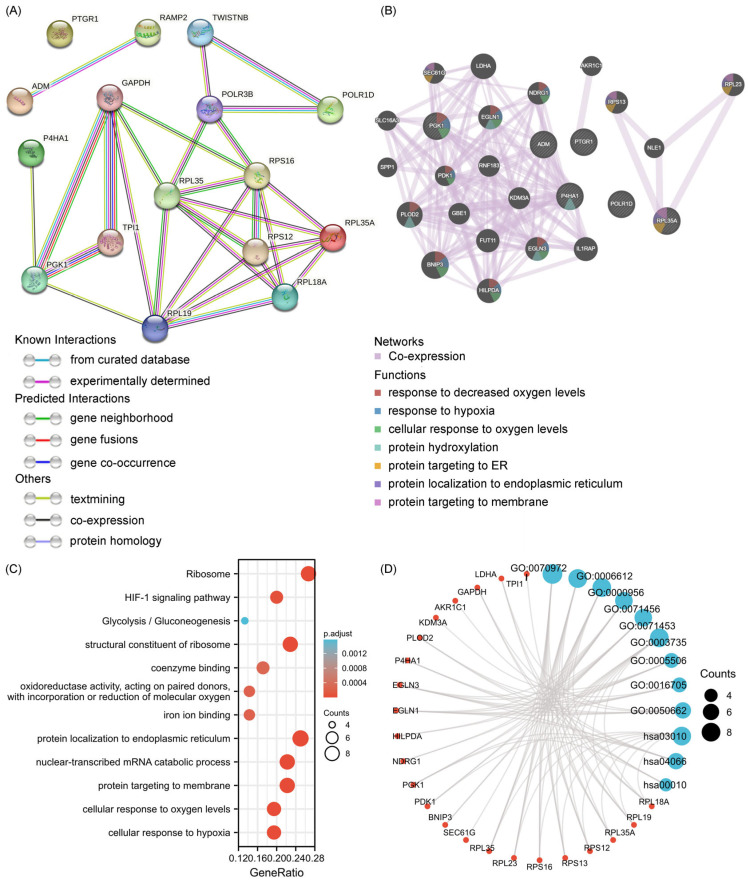
6-GPS co-expression network and functional enrichment. (**A**) PPI network and hub clustering genes of 6-GPS. (**B**) GeneMANIA demonstrates the gene interaction network of 6-GPS. (**C**) GO/KEGG enrichment analysis for 6-GPS. (**D**) Network visualization of GO/KEGG enrichment analysis for 6-GPS.

**Figure 8 ijms-25-01031-f008:**
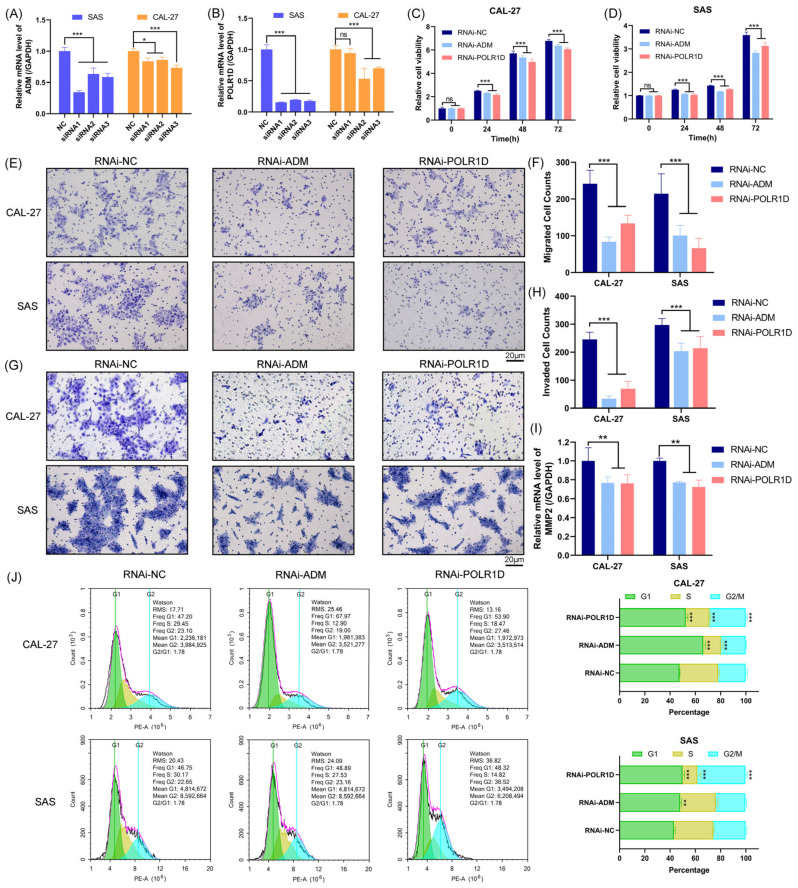
Correlations between *ADM*/*POLR1D* knockdown and biological characterizations of CAL-27 and SAS cells. The expression of *ADM* (**A**) and *POLR1D* (**B**) was interfered with using siRNAs. The interfering efficiency was examined by RT-qPCR. CCK-8 assay was conducted to detect the proliferation of CAL-27 (**C**) and SAS (**D**) cells after the targeted gene silence. Cell migration (**E**–**F**) and Invasion (**G**–**H**) were detected in CAL-27 and SAS cells transfected with RNAi-*ADM* or RNAi-*POLR1D* by Transwell assay. mRNA expression levels of *MMP2* in CAL-27 and SAS cells with the *ADM* or *POLR1D* knockdown (**I**). Cell-cycle analysis demonstrated an increase in the proportions of the G1 phase upon knockdown of *ADM* and *POLR1D* (**J**). Data are shown as mean and SD of triplicates (mean ± SD). (*: *p* < 0.05; **: *p* < 0.01; ***: *p* < 0.001; ns: No significance; vs. RNAi-NC).

**Figure 9 ijms-25-01031-f009:**
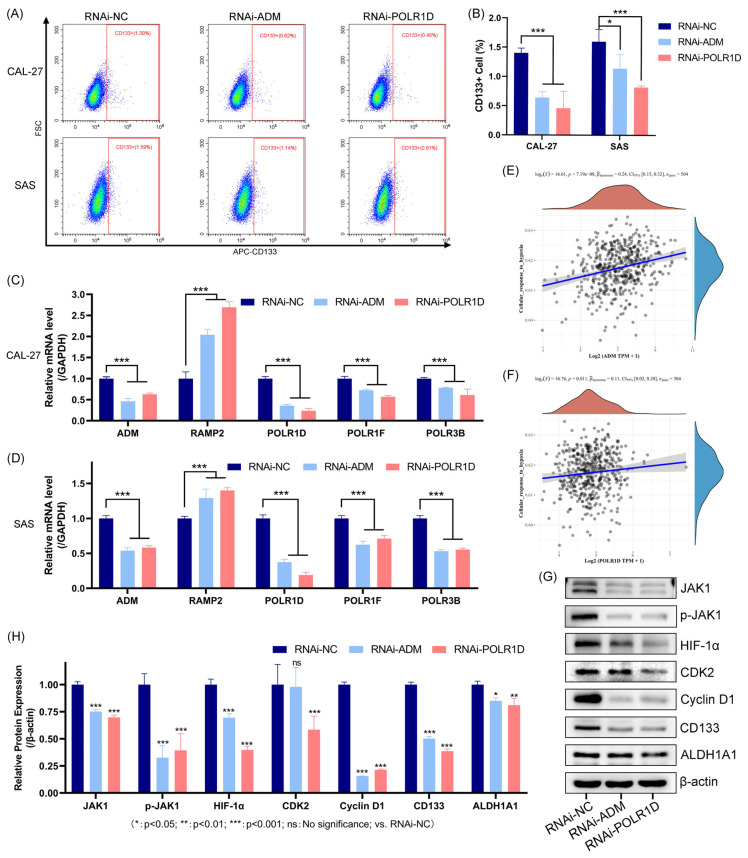
Mechanistic role of *ADM* and *POLR1D* in OSCC. Flow cytometry analysis of CD133 cell surface expression as a marker of cancer stem cells ((**A**) Flow cytometry images; (**B**) statistical analysis). The networking genes of *ADM* and *POLR1D* expression levels in CAL-27 (**C**) and SAS (**D**) cells were checked by RT-qPCR. Spearman’s correlation analysis between *ADM* (**E**) and *POLR1D* (**F**) and pathway scores. Expression of the protein in SAS cell was determined by Western blot ((**G**) Western blot images; (**H**) statistical analysis). Data are shown as mean and SD of triplicates (mean ± SD). (*: *p* < 0.05; **: *p* < 0.01; ***: *p* < 0.001; vs. RNAi-NC).

**Table 1 ijms-25-01031-t001:** The primer sequences of the target genes.

Genes	Forward Primer (5′-3′)	Reverse Primer (5′-3′)
*ADM*	CGAAAGAAGTGGAATAAGTGGGC	AGTTGTTCATGCTCTGGCGGT
*POLR1D*	AAAGAGGGCGATAAGGAACCAG	TTTCGTACTTGTCCTGGCTGC
*RAMP2*	ACTTTGCCAACTGCTCCCTG	GCCTCACTGTCTTTACTCCTCCAT
*POLR1F*	AGTGACTCCAGTGGTTACCAAAGT	GGTGAGCATTTCAAAGGTGGG
*POLR3B*	CATCCGCAATGCCTTACCT	CCCTTTTCTATCAGCCTCCAC
*MMP2*	CTCATCGCAGATGCCTGGAA	TTCAGGTAATAGGCACCCTTGAAGA
*GAPDH*	GGAAGCTTGTCATCAATGGAAATC	TGATGACCCTTTTGGCTCCC

## Data Availability

The results shown here are based on data generated by the ICGC database: https://dcc.icgc.org/releases/current/Projects (accessed on 3 November 2022), the GEO database: https://www.ncbi.nlm.nih.gov/geo (accessed on 17 October 2022) under the accession numbers GSE172577, and the HPA database: https://www.proteinatlas.org (accessed on 21 January 2023).

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
