# Peer review of "Identification and Validation of a Prognostic Signature Derived from the Cancer Stem Cells for Oral Squamous Cell Carcinoma"

_ijms, 2024, doi:10.3390/ijms25021031_

Round 1

Reviewer 1 Report (Previous Reviewer 2)

Comments and Suggestions for Authors

The revisions are satisfactory and the manuscript may be accepted.

Author Response

Response to Reviewer 1 Comments

1. Summary

    We would like to thank reviewer for the positive comment on our manuscript. And we have addressed your previous suggestion by including additional descriptions of the quantitative and statistical analysis for some of the experimental results in the Methods section of the manuscript. We hope this will enhance the clarity and rigor of our study.

2. Questions for General Evaluation

Reviewer’s Evaluation

Response and Revisions

Does the introduction provide sufficient background and include all relevant references?

Yes

Are all the cited references relevant to the research?

Yes

Is the research design appropriate?

Can be improved

Are the methods adequately described?

Can be improved

Are the results clearly presented?

Yes

Are the conclusions supported by the results?

Can be improved

3. Point-by-point response to Comments and Suggestions for Authors

Comments 1: The revisions are satisfactory and the manuscript may be accepted.

Response 1: Thank you for your positive feedback! We are glad to hear that the revisions were satisfactory and that the manuscript may be accepted. We appreciate the time and effort you have put into reviewing our work. If there are any further suggestions or revisions needed, please let us know.

Reviewer 2 Report (New Reviewer)

Comments and Suggestions for Authors

This study investigated the prognostic value of a panel of 6 genes (6-GPS) identified from 1344 cancer stem cells-related genes in OSCC. The authors validated their 6-GPS in an independent cohort of 40 OSCCs from ICGC. Two of the 6 genes were chosen (ADM and POLR1D) for subsequent functional investigation using two OSCC cell lines and shown that both genes were crucial for proliferation, migration and invasion of OSCC cells acting through the JAK/HIF-1 pathway.  Overall, the study appeared to be logical but experimental details on number of replicates were missing from most figures render their data risky. Below are several comments that need to be addressed:

11.       It was not clear why TACSTD2 and KRT19 were selected as representative stem cell genes for cell cluster analysis? What about the other well studied stem cell genes such as CD44, CD133, ALDH1A1, do these also show the same cell clustering as for TACSTD2 and KRT19 in Figure 1?

22.       It seems odd that the initial AUC values obtained from the test cohort for 6-GPS were below 0.7 (Figure 3G) and subsequently AUC=1 in the validation cohort (Figure 5E). Please explain this discrepancy.

33.     It is not clear if Figure 6A and B were derived from the same dataset or different? Why the statistical significance were different for PTGR1 and RPL35A? Figure legend should explain clearly what these two figures represent in details.

44.     Figure 8A and B seems swapped according to the figure legend, A is for ADM and B for POLR1D? Same for 8C and 8D.

55.       Figure 8: Insufficient details were provided in the figure legend to indicate the number of replicates performed in all the experiments in this figure. The n number of technical repeats and biological repeats should be clearly stated for each figure. For example what does each bar represent in 8A? Was it a mean of n=3 independent experiments? For Figure 8J, how many independent FACS were performed (indicate the n number in figure legend) to enable statistical analysis?

66.       Figure 9 – also lacks information on the number of repeats for each experiment. How was Figure 9H measured, from how many western blot repeats to obtain a mean value?

77.       Methods - Insufficient details were given for the section on RT-qPCR - not clear what reagent was used to quantify mRNA in the qPCR system and what cycling profile was used. The authors should provide sufficient information to enable independent replication of their data.

Comments on the Quality of English Language

No problem

Round 2

Reviewer 2 Report (New Reviewer)

Comments and Suggestions for Authors

The authors have sufficiently answered all my queries.

This manuscript is a resubmission of an earlier submission. The following is a list of the peer review reports and author responses from that submission.

Round 1

Reviewer 1 Report

Comments and Suggestions for Authors

In this review, the authors investigated a new prognostic feature associated with tumor stem cell marker genes to improve the management of patients with oral squamous cell carcinoma (OSCC).

Minor revisions need to be done before acceptance.

1)  General Revision:

- Typography: the authors should read thoroughly their manuscript and check: 1) space between words; 2) English of some sentences.

2)  Introduction section:

- I suggest that further research progress regarding recent cancer stem cells (CSC) markers in OSCC need to be described more.

3) Materials and Methods section:

- Please specify the sex of the patients and further explain whether there might be gender differences in stem cell markers in OSCC patients.

3)  Discussion section:

- Since only the biological functions of ADM and POLR1D in the field of OSCC were investigated in this study, it is necessary to mention previous studies that have clarified the role of the other 4 genes within 6-GPS in OSCC.

Comments on the Quality of English Language

English language to be revised

Reviewer 2 Report

Comments and Suggestions for Authors

Shi et al. try to identify prognostic signatures for oral squamous cell carcinoma. The authors mainly derived their main idea from publically available transcriptomic data and supported it with some laboratory experimental data. The topic is relevant to stem cell research as this is essential information about the prognostic signatures in OSCC. The research question needs to be addressed better in this manuscript, and there must be more gaps in proving their findings.

  1. The title states about cancer stem cells. The publically available GEO dataset (GSE172577) deals with stem cells; this dataset talks about single-cell profiling of tumor-infiltrating TCF7+ T cells in oral cancer. The authors analyzed their results from GEO on TCGA data and OSCC epithelial cell lines. In vitro validation of the prognostic signatures in stem cells needs to be included.

  2. Since the actual GEO dataset deals with tumor-infiltrating TCF7+ T cells, validating infiltration or stem cells markers is essential. 

  3. Figure 7 shows networking analysis. The networking genes of ADM and POLR1D can be checked by qPCR. 

  4. GO enrichment analysis is shown in figure 7C. The top GOs are protein targeting to the cell membrane, nuclear-transcribed mRNA catabolism, cellular response to oxygen levels, cellular response to hypoxia, and protein localization to the ER. Hypoxia must play a role here. Hence, I recommend authors to their cells in hypoxia and study the ADM/POLR1D knockdown and biological characterizations. 

  5. No detailed mechanism has been established with molecular-level evidence. The two key genes' interactions and other immune genes or cancer-specific pathways should be addressed experimentally. 

  6. Using CIBERSORT, the interaction of ADM/POLR1D with other tumor-infiltration markers in OSCC should be studied.

  7. A graphical representation of overall results would be advantageous.